# Can DFT Calculations Provide Useful Information for SERS Applications?

**DOI:** 10.3390/molecules28020573

**Published:** 2023-01-06

**Authors:** Maurizio Muniz-Miranda, Francesco Muniz-Miranda, Maria Cristina Menziani, Alfonso Pedone

**Affiliations:** 1Dipartimento di Chimica “Ugo Schiff”, Università degli Studi di Firenze, Via Lastruccia 3, 50019 Sesto Fiorentino, Italy; 2Dipartimento di Scienze Chimiche e Geologiche, Università degli Studi di Modena e Reggio Emilia, Via Campi 103, 41125 Modena, Italy

**Keywords:** SERS spectroscopy, applications, DFT calculations, metal surface modeling

## Abstract

Density functional theory (DFT) calculations allow us to reproduce the SERS (surface-enhanced Raman scattering) spectra of molecules adsorbed on nanostructured metal surfaces and extract the most information this spectroscopy is potentially able to provide. The latter point mainly concerns the anchoring mechanism and the bond strength between molecule and metal as well as the structural and electronic modifications of the adsorbed molecule. These findings are of fundamental importance for the application of this spectroscopic technique. This review presents and discusses some SERS–DFT studies carried out in Italy as a collaboration between the universities of Modena and Reggio-Emilia and of Florence, giving an overview of the information that we can extract with a combination of experimental SERS spectra and DFT modeling. In addition, a selection of the most recent studies and advancements on the DFT approach to SERS spectroscopy is reported with commentary.

## 1. Introduction

The SERS (surface enhanced Raman scattering) [1,2,3] effect is able to hugely amplify the Raman response of molecules adhering to the nanostructured surfaces of metals such as silver, gold, and copper. This effect, which usually produces a Raman signal over one million times larger than that observed in nonadsorbed molecules, is generally considered to be the product of two contributions, one electromagnetic and another chemical. The first contribution must satisfy two requirements: (i) the substrate must be composed of a metal with high optical reflectivity, that is, the imaginary part of the dielectric constant must be very small in the spectral region of Raman excitation; (ii) the wavelength of the exciting radiation must fall within the plasmon band of metal nanoparticles so that the molecules adhering to metal undergo an electric field much larger than that at long distances from the surface. This contribution, providing Raman enhancement factors up to 10^7^, allows observation of the spectra of molecular submonolayers adsorbed on nanostructured metals and does not require the formation of chemical bonds between molecules and metal. The second contribution to the SERS enhancement, which is observed when the molecule is chemisorbed on active sites of the metal surface, depends on the change of the molecular polarizability due to the formation of metal–molecule complexes. The electromagnetic contribution is essential to the SERS effect, but the chemical one, even if it provides Raman enhancement factors up to only 100, is important in determining the spectral pattern, as the formation of surface complexes produces significant frequency-shifts and intensity changes in the SERS bands with respect to those observed in the normal Raman spectra of nonadsorbed molecules.

In this respect, the density functional theory (DFT) approach can be used for interpreting the SERS spectra of chemisorbed molecules. In fact, these DFT calculations are based on complexes constituted by molecular ligands and active sites of the metal surface, which are modeled as adatoms or adclusters with one or a few atoms, considered almost isolated on the metal surface. This computational approach allows us to obtain very important results: (1) the spectral positions of the SERS bands and the frequency shifts with respect to the normal Raman bands of the nonadsorbed molecules; (2) the relative intensities of the observed SERS bands; (3) the identification of the molecular sites directly involved in the interactions with the metal substrates and, consequently, also the adsorption geometries; (4) information on the nature of the surface’s active sites involved in the chemical interaction with the ligand molecules.

In recent decades, SERS spectroscopy has proved to be of fundamental importance in applications in a variety of research fields, including in chemical sensors, materials science, and biomedicine [4,5]. For all of these applications, it is essential to have detailed information on the molecule–metal interaction. The combined SERS–DFT approach makes it possible to achieve all of this information as illustrated above. In this regard, this review presents some of the computational studies carried out in Italy in the Department of Chemical and Geological Sciences of the University of Modena and Reggio-Emilia on the experimental results obtained in the Chemistry Department of the University of Florence. Finally, a selection of the most recent studies by different authors on the DFT approach to SERS spectroscopy is commented on and related to their applications.

## 2. Results

### 2.1. Xanthine Adsorbed on Silver Colloidal Nanoparticles

Xanthine, a nucleobase that is produced in the human body from adenine and guanine (Figure 1), can provoke diseases similarly to uric acid when it is deposited at abnormal levels [6]. The latter can be detected with SERS spectroscopy in diagnostic procedures in order to prevent metabolic dysfunctions. In particular, FT-SERS were obtained in the near-infrared spectral region [7], where interference due to fluorescence does not occur, and analyzed by DFT calculations.

The DFT computational approach allowed identification of the “marker” SERS bands of xanthine by considering the molecule linked to silver particles, as well as the molecular group involved in the interaction with the metal, by comparing the observed SERS spectrum with those calculated with all possible complexes (Figure 2). In practice, this computational study was complicated by the possible presence of two tautomers and of different complexes formed by molecule–metal interactions. Only the N(9)H-A complex satisfactorily reproduced both positions and relative intensities of the SERS bands, in particular the dominance of the SERS band around 650 cm^−1^ and the occurrence of the strong SERS band around 1700 cm^−1^. The modeling of the silver surface was efficiently performed by one positivized silver atom strongly interacting with the N7 atom together with the vicinal carbonyl group.

The use of DFT calculations is quite important because the identification of the marker bands of xanthine can be impaired by the presence of other biological components. This SERS–DFT investigation made it possible to use FT-SERS spectroscopy for biosensory and diagnostic purposes in body fluids in order to detect abnormal xanthine levels and prevent metabolic diseases.

### 2.2. Tapazole Adsorbed on Gold Colloidal Nanoparticles

The use of metal nanoparticles linked to biomolecules can be very advantageous in a therapeutic practice thanks to three possible applications: (i) to enhance the therapeutic effect of the drug; (ii) to allow efficient drug delivery due to the carrier action of the metal nanoparticles, which can release the drug where it is needed; (iii) to increase the therapeutic retention time in circulation, allowing the use of a reduced dose of the drug with limited side effects. Among metallic nanoparticles, those made of gold provide the best biomedical use, owing to their low toxicity in comparison with other noble metals and the absence of alterations due to oxidative processes.

This therapeutic strategy is particularly interesting in the case of a molecule such as methimazole, commercially known as tapazole (TZ), which has long been used for the treatment of hyperthyroidism. However, a long period of antithyroid therapy with tapazole can cause serious health consequences, including the onset of diseases such as vasculitis, lupus erythematosus, nephritis, and thrombocytopenia [8,9,10]. Therefore, gold colloidal nanoparticles functionalized with TZ allow a reduced dose of the drug with limited side effects. For this purpose, the study of the molecule–metal interaction, which was carried out by means of a combined SERS–DFT approach [11], is of great importance. In practice, the interpretation of the SERS data based on DFT calculations allowed identification of the molecular species, the thiolate anion (TZA), which chemically interacted with the gold surface only through the sulfur atom. The active sites of the metal surface were modeled as zero-charge gold adatoms or adclusters (Figure 3). In particular, the adoption of adclusters formed by three gold atoms instead of one-atom adatom did not provide a significant improvement in the comparison between simulated and experimental SERS spectra, thus confirming that the SERS profile of a chemisorbed molecule is determined by the local interaction between the molecular site and the active site of the surface. In the case of TZ, the molecular site of interaction was only the sulfur atom, as shown in Figure 4, where the Au-S stretching band observed at 202 cm^−1^ was calculated at 199 cm^−1^. This result can play an important role in the use of these nanohybrids in drug delivery because the Au–S bonds of organothiols adsorbed on gold nanoparticles underwent breaking with iodide anions. In such a way, TZ can be released right where it is needed as antithyroid drug.

### 2.3. Zeatin Adsorbed on Gold Colloidal Nanoparticles

Trans-zeatin (hereafter zeatin) is a natural purine derivative used in vitro in plant culture to stimulate growth and bud formation, whereas the cis form is biologically inactive [12]. In this respect, gold nanoparticles can act as vectors of zeatin into their living cells, where this drug can then be released by the effect of glutathione, an antioxidant present in all living cells, due to its strong affinity with metallic gold. On the other hand, Au nanoparticles have been shown to enter and accumulate in plant cells with very little toxicity [13,14]. To test the ability of gold nanoparticles to adsorb and transport zeatin, SERS experiments were performed for different possible complexes with gold [15] and combined with DFT calculations.

Two zeatin tautomers (Figure 5), N7(H) and N9(H), were considered linked to gold adclusters, neutral or positively charged. The best agreement between SERS data and Au complexes was obtained in the case of the N7(H) tautomer bound to gold through two different nitrogen atoms (Figure 6).

### 2.4. Corrosion Inhibitors on Copper Rough Surfaces

Copper has always been widely used for industrial purposes by virtue of its great thermal and electrical conductivity. However, it undergoes corrosion phenomena, especially in the presence of water [16], which seriously compromise its use. A solution to this problem is to protect the copper surfaces with anticorrosive films, thanks to the chemisorption of organic inhibitors, containing sulfur or nitrogen atoms. Among these corrosion inhibitors, azole derivatives such as benzotriazole [17] and 1,2,4-triazole [18] have been widely used. The latter, unlike the former, can be considered an environmentally friendly agent. To obtain information on the adsorption of 1,2,4-triazole on copper, we used SERS spectroscopy, which required roughening the metal surface, combined with a DFT study [19]. The computational approach was complicated by the existence of two different tautomers of the title molecule, indicated as 1H and 4H, based on the presence of a hydrogen atom linked to the nitrogen atom N1 or N4, respectively, and by the fact that the bond with metal could occur via two different molecular sites, the sp2-type nitrogen atoms N2 and N4 (see Figure 7). Furthermore, in the case of the 4H tautomer, bonding with two copper adatoms was possible.

The modeling of the metal surface was carried out with single copper atoms, neutral or positively charged, given the oxidation facility of the copper surface. A reasonable agreement between observed and simulated SERS spectra was obtained only in the case of the 1H tautomer bonded to a Cu+ ion by the nitrogen atom N4. This adsorption allowed forming chains of adsorbed molecules linked together by hydrogen bonds, thus creating a compact layer of triazole that can impair metal corrosion (see Figure 8).

A confirmation of this was shown by observing the SERS spectrum of 1,2,4-triazole adsorbed on copper, where the bands at about 530 and 620 cm^−1^, due to the formation of a Cu(I) multilayer [20], were completely absent. On the contrary, these bands, indicative of an initial stage of corrosion of copper, were observed in the SERS spectrum of imidazole, an azole structurally very similar to triazole but unable to bind the adsorbed molecules together (Figure 9).

### 2.5. Push-Pull Molecule Adsorbed on Silver Colloidal Nanoparticles

Push-pull chromophores exhibit two end groups with electron-donor and electron-acceptor characters connected by a π-conjugated bridge, resulting in a large molecular dipole moment. These molecules find important nonlinear optical (NLO) applications; in particular, 4-dimethylamino-4-nitrostilbene (DANS) [21,22,23] is widely exploited in organic light-emitting diodes (OLEDs), in bioimaging, and in radiation therapy applications. The peculiar behavior of push-pull molecules is often described by a two-state model with a neutral form predominant in the ground state and a zwitterionic form predominant in the lowest excited state, as shown in Figure 10 for DANS.

However, the surrounding environment modifies their properties, for example, with different polarity solvents or the chemical interaction with a metal surface. In order to improve the NLO properties of DANS, we tested the chemical and physical behavior of this molecule when adsorbed on colloidal silver with SERS measurements and DFT calculations [24]. DANS strongly absorbed in the region of the green-blue radiation [25], as also predicted by the TD-DFT calculations (see Figure 11). Therefore, a Raman resonance effect was expected in addition to SERS enhancement.

What was not expected, however, was the progressive appearance of strong bands at 1136, 1400, 1444, and 1590 cm^−1^ not attributable to the SERS spectrum of the molecule (see Figure 12). This could be explained by a photoreaction of the molecules adsorbed on the silver particles. In fact, this effect as observed in silver colloid was not observed in the solution.

This plasmon-induced reaction could be considered due to the catalyzing effect of the nanostructured surface of the colloidal silver particles. The DFT approach was very important in understanding the adsorption of DANS on silver and to clarify the type and mechanism of this photoreaction. By considering the molecule linked through the nitrogroup to a silver adcluster, Ag_3_^+^, good agreement was found between the observed and simulated SERS spectra (see Figure 13).

The DFT calculations also showed that the interaction with silver modified the charge distribution in the molecule, especially by accumulating negative charge on the oxygen atoms of the nitrogroup, whereas the methyl groups became more positive. In other words, the DANS/Ag complex acquired a larger quinonoid character along with a strong decrease of the HOMO–LUMO gap (1.5725 eV, instead of 2.8931 eV). In this situation, a photoreduction of nitro group to N=N group (azo group) was possible due to the effect of the visible blue-green radiation, thus forming an azo-derivative as shown in Figure 14.

Confirmation of this was obtained thanks to the DFT calculations, which showed in the simulated Raman spectrum of the azo-derivative (Figure 15) precisely the strong “spurious” bands observed in Figure 12.

This SERS–DFT investigation clarified the adsorption of DANS on silver nanoparticles, suggesting a significant increase of the nonlinear properties when this molecule was linked to silver, with interesting applications. Moreover, the DFT approach explained the mechanism of photoreduction under green-blue irradiation. This reaction could be important for the opto-response of the adsorbed molecules, limiting their potential applications.

## 3. Computational Details

All DFT calculations of the molecule–metal complexes presented here were performed with the Gaussian 09 suite of programs [26] or previous editions (Gaussian 98, Gaussian 03), the B3LYP [27,28] hybrid exchange and correlation functional, and the Lanl2dz basis set [29,30,31]. The latter consists of the Dunning–Huzinaga full double zeta [30] on first-row atoms and of the Los Alamos pseudopotential for core electrons plus a double-zeta basis for valence electrons. In order to better reproduce the positions of the observed Raman bands, different basis sets were also employed: 6-31G(d), 6-31++G(d,p), or 6-311++G(d,p) basis sets [32,33] for all atoms except silver and gold, which were instead described by the Lanl2dz basis set. With mixed basis sets, a 0.98 factor was generally adopted for uniformly scaling the calculated harmonic frequencies. These, as well as the optimized structures, were computed while adopting tight convergence criteria. By allowing all parameters to relax, the optimized geometries corresponded to true energy minima as revealed by the lack of imaginary values.

## 4. Recent Developments

The DFT approach is increasingly used for SERS spectroscopy by several research groups, as demonstrated by the most recent studies [34,35,36,37,38,39,40,41,42,43,44,45,46,47,48,49,50,51,52,53,54,55,56,57,58,59,60,61,62,63,64] in which close links between DFT calculations and SERS applications are shown. Among these studies, the following studies devoted to applications in sensors, especially for environmental protection, and to applications in biomedicine, are worthy of particular attention. In this section, we commentate on this selection of recent SERS–DFT investigations.

### 4.1. Sensors

Formaldehyde is a highly water-soluble aldehyde, which can sterilize and prevent bacterial reproduction. However, formaldehyde has been recognized by the World Health Organization as toxic, carcinogenic, and teratogenic. Therefore, it is necessary to develop an efficient analytical method for detecting trace formaldehyde in food samples. Colloidal gold nanoparticles were used as SERS substrates [43] to achieve rapid determinations of formaldehyde from rice flour and duck blood products. Density functional theory (DFT) calculations were performed to recognize the characteristic peaks and vibrational modes of formaldehyde in the presence of other products.

Pesticides play an important role in modern agriculture, but excess pesticides accumulated in soil, water, plants, crops, food, and beverages represent a serious problem for human health. Hence, a sensitive and reproducible detection method becomes an essential requirement for their use. SERS investigation by adsorption of analyte on silver substrate was adopted in this paper [47] to monitor the levels of pesticides such as chlorpyrifos (*O*,*O*-diethyl *O*-3,5,6-trichloropyridin-2-yl phosphorothioate), which is widely applied to protect crops and seeds from insects and worms. However, SERS spectra of chemical pesticides are generally complicated by the presence of multiple atoms (Cl, F, P, S, N, O, etc.) able to link silver colloidal nanoparticles. This involves serious difficulties in determining the characteristic SERS peaks for each type of compound. In this regard, the DFT approach adopted here allowed identifying the marker bands of this pesticide for its rapid screening in the environment.

This work [55] presented a SERS and DFT study of fipronil adsorbed on colloidal silver nanoparticles. Fipronil (5-amino-1-[2,6-dichloro-4-(trifluoromethyl) phenyl]-4-[(trifluoromethyl)-sulfinyl]-1*H*-pyrazole-3-carbonitrile) is a potent insecticide widely used in agriculture to control pests with high efficiency at very low concentrations. However, fipronil in food is known to cause serious diseases in the human body by provoking changes in the blood biochemistry. The characteristic peaks of the SERS spectrum of fipronil adsorbed on silver nanoparticles were identified here at very low concentrations with the help of DFT calculations in food as well as in the water and soil environment.

Silver nanodendrites were utilized [58] for the ultrasensitive identification of explosive molecules, such as 2,4-DNT (2,4-dinitrotoluene), and pesticides, such as Thiram (dithioperoxyanhydride), by using SERS spectroscopy. The DFT calculations were performed to assign the Raman bands of the analyte molecules, and their results compared with experimental results with reasonably good agreement. This enabled the simultaneous detection of various explosives and pesticides from a complex mixture of molecules.

Nitrobenzene and aniline are the main representatives of nitro or amino compounds of benzene, which are generally toxic. Thus, they have become important indicators of environmental pollution. The current determination methods of aniline and nitrobenzene, such as gas or liquid chromatography and spectrophotometry, require long analysis time and pretreatment. SERS spectroscopy avoids these disadvantages, but clear and sure identification and assignment of the observed bands is needed; it was obtained thanks to DFT calculations [60]. Moreover, this approach allowed a better understanding of the interaction between molecules and gold substrates, thus providing theoretical support for applying the SERS technique in high-sensitivity detection of environmental pollutants.

Bufotenine (5-hydroxy-N,N-dimethyltryptamine), hereafter BUF, is a natural tryptamine derivative with prominent hallucinogenic activity. In this paper [63], surface-enhanced Raman scattering (SERS) was employed by using silver nanoparticles for the detection of BUF at trace amounts. The vibrational characteristics of this molecule were identified by DFT simulations. This combined SERS–DFT approach allowed a rapid and accurate sensing for the identification of BUF and could represent a method of detection and characterization of other substances of forensic, criminological, and social importance.

### 4.2. Biomedicine

Among the most commonly used anesthetic drugs, procaine (2-diethylaminoethyl-p-aminobenzoate) is also used in the treatment of depression, obesity, inflammation, and cancer. At higher doses, however, procaine can lead to lethal outcomes, including causing cardiac arrest. Furthermore, the presence of procaine in pharmaceutical preparations, even in minimal doses, has given rise to several allergenic cases. For this reason, trace detection of procaine adsorbed on silver nanoparticles was performed with SERS spectroscopy and combined with DFT calculations [35]. This approach was capable of providing certain attribution of the SERS bands along with information on the type of adsorption on the metal surface, which was ascertained to occur via the NH_2_ group. This study is important for limiting the risks associated with the exposure to variable dose concentrations of the drug.

α-lipoic acid (hereafter LA) is a biomolecule involved in the metabolism of glycine and serine, thus the study of LA is potentially able to provide many benefits for biomedical applications regarding metabolic diseases and cancerous phenomena. A silver surface can be considered an artificial biological interface and the adsorption of a molecule on it, thus representing a model for the adsorption process on membranes or other biological surfaces. In order to understand the action of LA, it is essential to identify any alteration of the adsorbed species relative to the structure of the free molecule. For this reason, SERS spectroscopy of LA adsorbed on Ag colloidal nanoparticles was adopted in combination with DFT calculations [36]. An LA molecule was chemisorbed on the Ag nanoparticle surface through the carboxyl group in predominantly perpendicular orientation with respect to the silver surface. This study can represent a basis for important medical or therapeutic treatments of this biologically active molecule.

Folic acid (hereafter FA), a well-known vitamin essential for the production and maintenance of DNA and RNA, was extensively used in drug delivery systems as a targeting biomolecule. In this way, the study of the SERS spectral pattern of FA adsorbed on gold nanoparticles [42] can be useful for understanding its adsorption mechanism, which is necessary for safely using this molecule–metal hybrid as a nanocarrier. In this regard, DFT calculations allowed deducing how chemical interactions occurred between FA and the gold surface. FA interacted with the gold surface through the pteridine moiety in a tilted geometry.

Clotrimazole (hereafter CTZ), a common derivative of imidazole, has been used to treat fungal and yeast infections affecting sensitive skin areas. In particular, it was proven effective in treating candidiasis, a fungal disease caused by the pathogen *Candida albicans*, by inhibiting the proteins in the fungal cell membrane. Many research groups, prompted by the increasing use of CTZ as an efficient drug, have been engaged in studying the structural and spectroscopic properties of CTZ as well as finding efficient trace-level detection techniques. In this regard, SERS spectroscopy is particularly suitable also thanks to the presence of electronegative atoms as nitrogen and chlorine in the molecule, which are capable of chemically bonding to a metal surface. In this paper [45], CTZ was adsorbed on silver nanoparticles loaded on graphene and studied with SERS spectroscopy with the aid of DFT calculations. The latter played an important role in the identification of the marker’s SERS bands and those of the conformer linked to metal, and it provided insight into the mechanism of CTZ’s action in treating candidiasis.

In this work [46], a SERS–DFT study was performed on the neurotransmitter dopamine. Levels of dopamine indicate the neurological status of a human being, playing a central role in diseases such as Parkinson’s or schizophrenia. SERS represents a powerful tool for detecting dopamine levels thanks to its specificity and sensitivity. SERS substrates were fabricated in this study by sintering silver nanoparticle paste onto a fused silica substrate using a femtosecond laser. DFT calculations for different models of silver clusters linked to dopamine allowed identifying the signature SERS bands of this neurotransmitter and obtained information on the charge transfer process between molecule and metal. This approach could allow performance of a quantitative analysis of dopamine in clinical samples, even in the presence of multiple neurotransmitters.

The high disinfectant activity of benzalkonium chloride (BAC), a cationic surfactant quaternary ammonium compound, has been increasingly used, in particular for inactivating the SARS-CoV-2 virus, which causes COVID-19. However, because BAC potentially poses risks for human and environmental safety, it is necessary to know the benefits versus the risks of a very extensive use of BAC use in commercial products. Hence, in this paper [59], an efficient and sensitive SERS method to rapidly detect BAC adsorbed on glass surfaces was employed by using AgNP nanostructures. In this regard, DFT calculations lent valid support to the SERS technique for detecting very low amounts of BAC along with providing significant information on the properties of BAC and its adsorption on metal. This approach constituted a valid model for investigating the presence of disinfectant agents similar to BAC on various surfaces such as textiles, aluminum, wood, and stainless steel.

## 5. Conclusions

The examples reported in this review show that the DFT calculations carried out to support the SERS spectra, albeit relatively simple in their execution, are not mere simulation exercises. Indeed, they allow extracting the most information that SERS spectroscopy is potentially able to provide. The information obtained on the adsorption on metal nanoparticles, on the chemical species linked to metal, on the molecular sites directly involved in the chemical interaction, and on the structural and electronic modifications of the adsorbed molecule are of fundamental importance in view of the present and future applications of this spectroscopic technique. In this respect, the most recent SERS investigations in the field of medical diagnosis are of particular relevance [65,66,67,68,69].

## Figures and Tables

**Figure 1 molecules-28-00573-f001:**
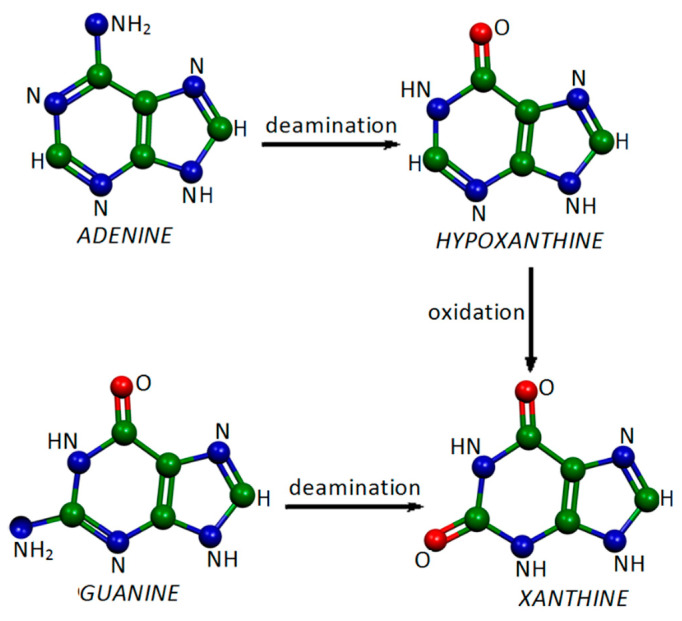
Formation of xanthine from adenine and guanine.

**Figure 2 molecules-28-00573-f002:**
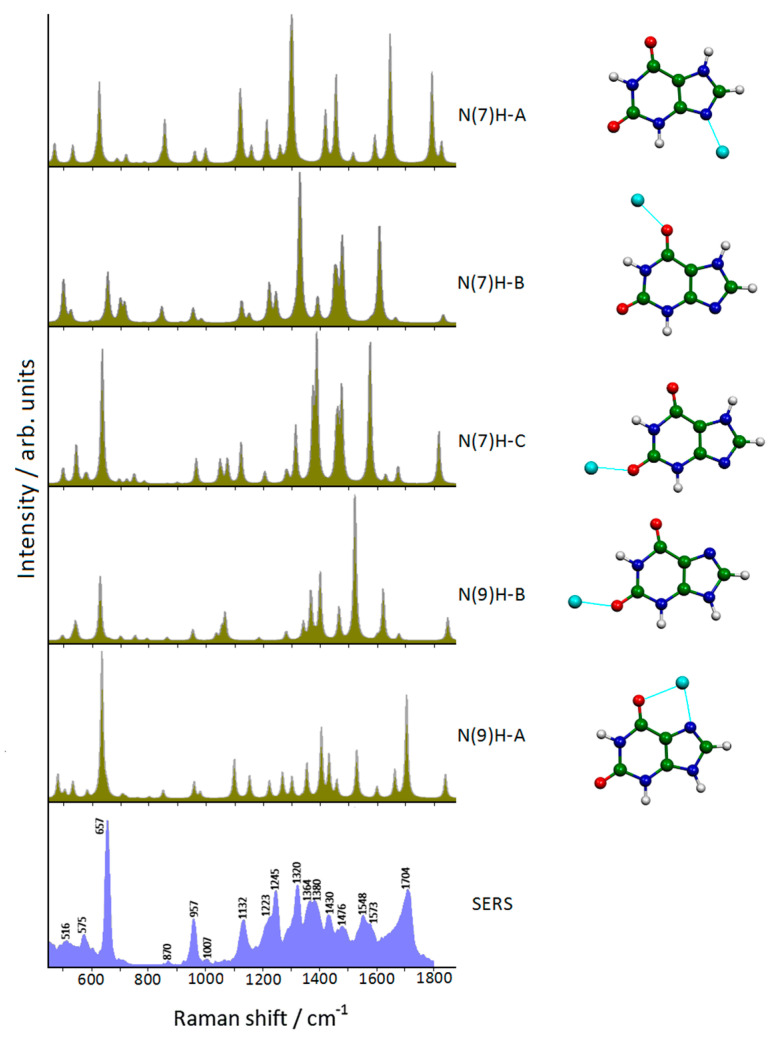
DFT-simulated Raman spectra of xanthine–silver complexes compared with the observed SERS spectrum of xanthine adsorbed on Ag colloidal nanoparticles (1064 nm laser excitation).

**Figure 3 molecules-28-00573-f003:**
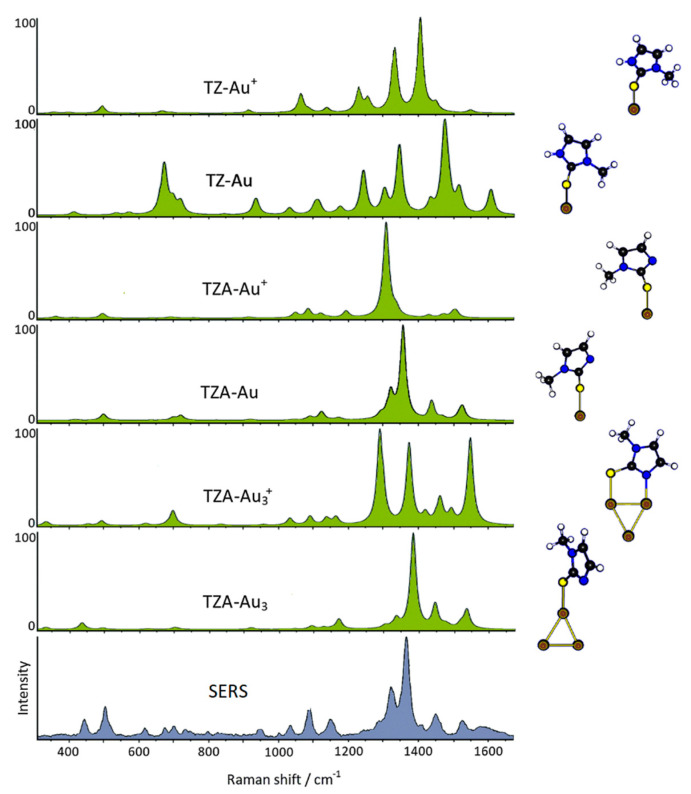
DFT-simulated Raman spectra of tapazole–gold complexes compared with the observed SERS spectrum of tapazole adsorbed on gold colloidal nanoparticles (647.1 nm laser excitation).

**Figure 4 molecules-28-00573-f004:**
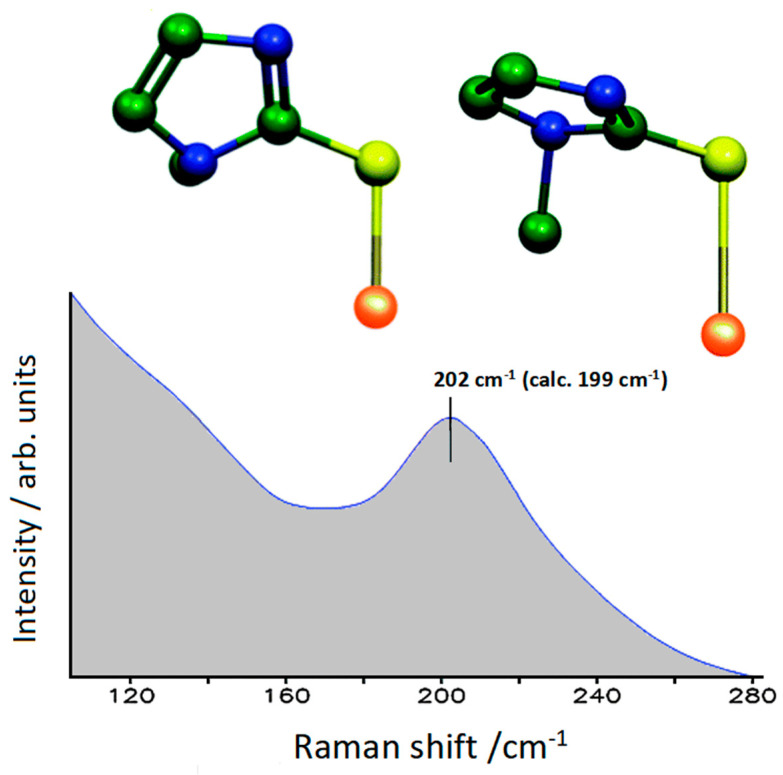
Low-frequency SERS spectra (exc. 1064 nm) of TZ in Au colloids. The normal mode calculated at 199 cm^−1^ for the TZA-Au model is shown above for the two maximum deformations (hydrogen atoms hidden).

**Figure 5 molecules-28-00573-f005:**
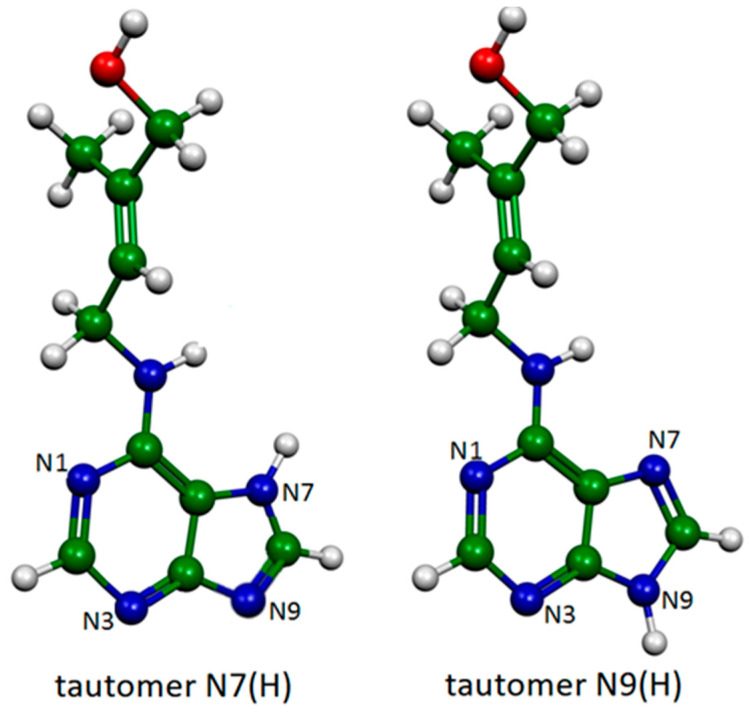
Tautomeric forms of trans-zeatin.

**Figure 6 molecules-28-00573-f006:**
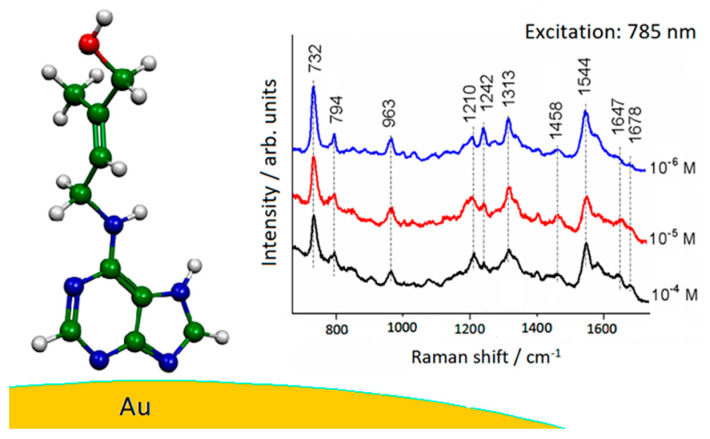
Adsorption of zeatin on a gold nanoparticle along with the SERS spectra obtained in Au colloid with different concentration.

**Figure 7 molecules-28-00573-f007:**
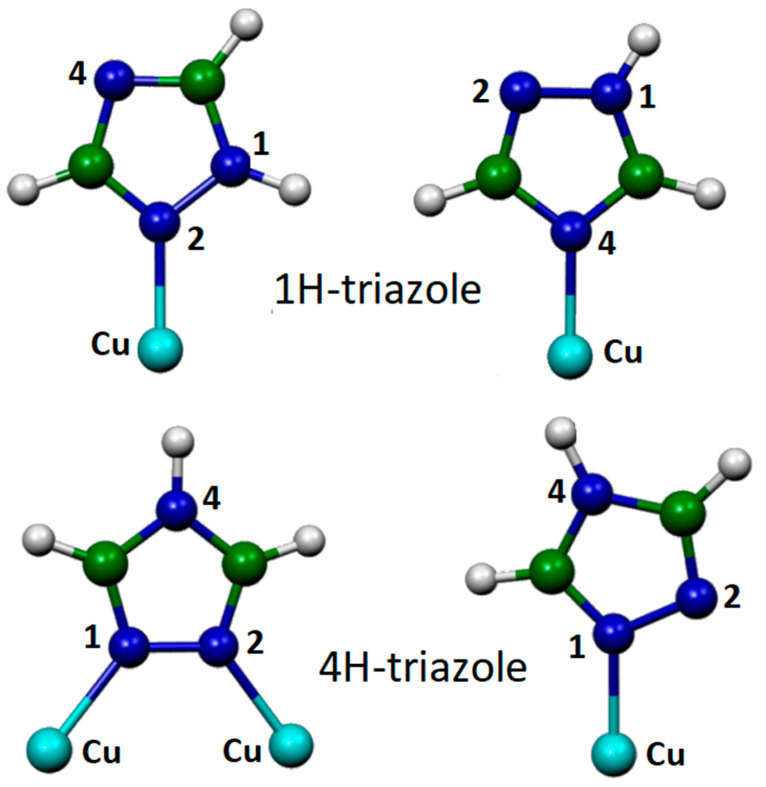
Model systems for 1*H*/copper and 4*H*/copper (lower panel) complexes.

**Figure 8 molecules-28-00573-f008:**
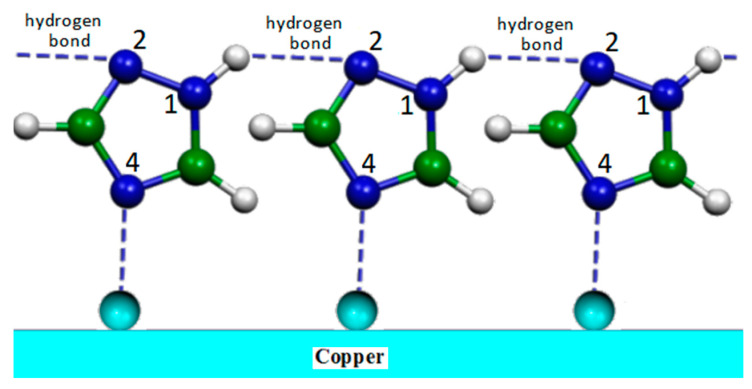
Adsorption model of the 1,2,4-triazole molecule on the copper substrate.

**Figure 9 molecules-28-00573-f009:**
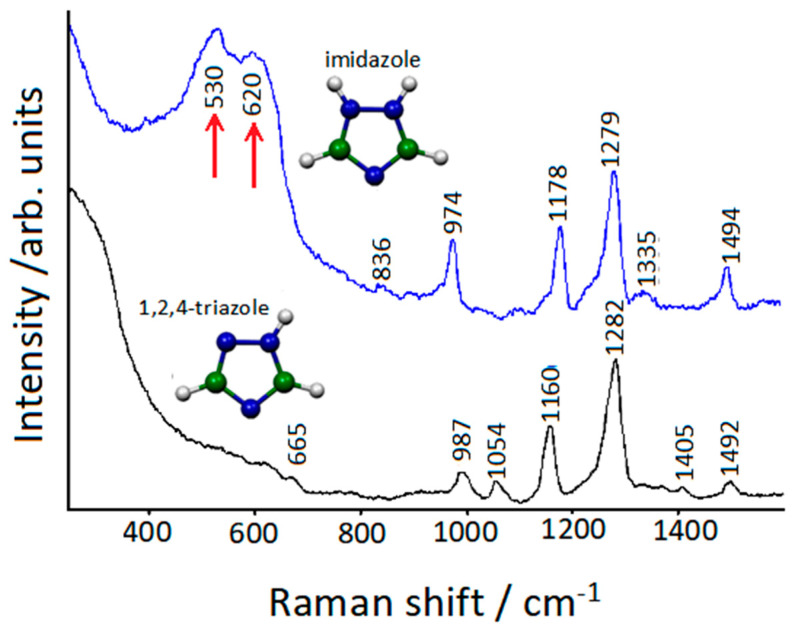
SERS spectra of 1,2,4-triazole and imidazole absorbed on etched copper surfaces.

**Figure 10 molecules-28-00573-f010:**
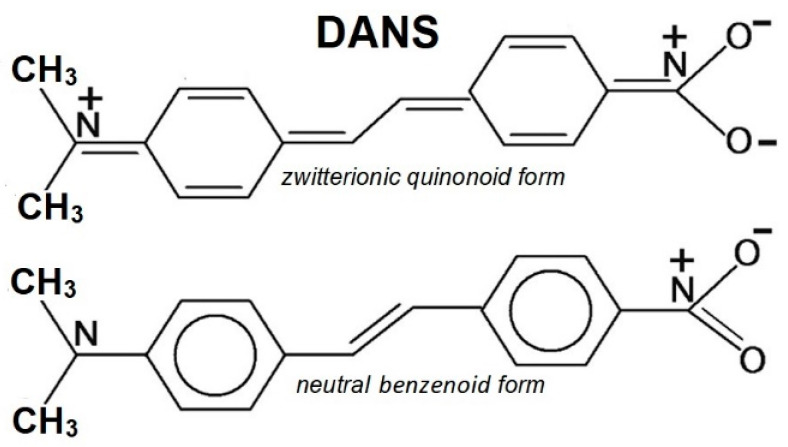
Resonance structures of DANS.

**Figure 11 molecules-28-00573-f011:**
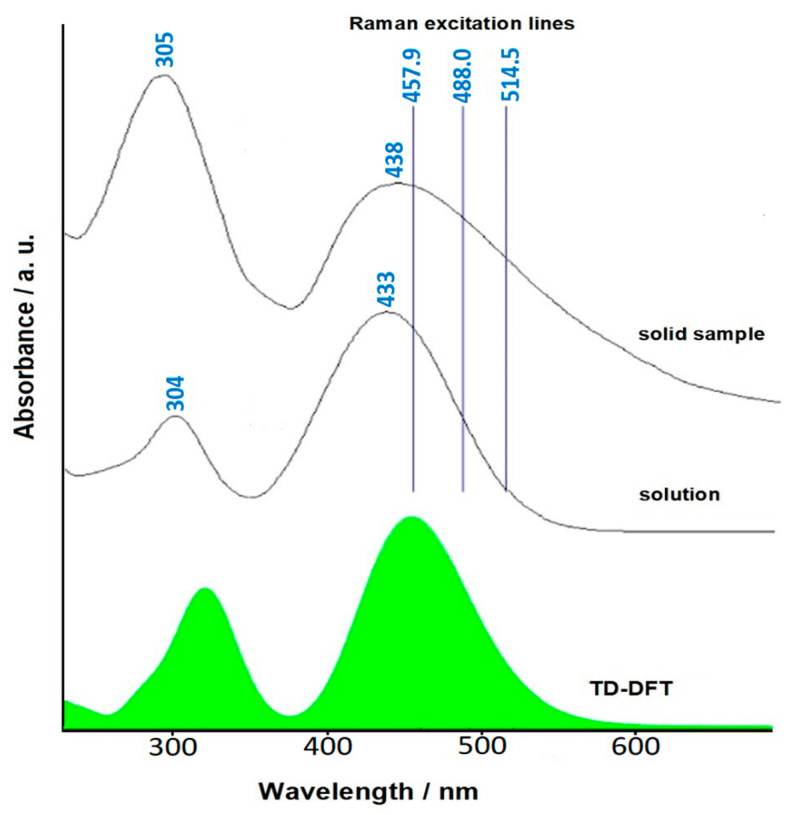
UV–vis absorption spectra of DANS along with the simulated TD-DFT spectrum.

**Figure 12 molecules-28-00573-f012:**
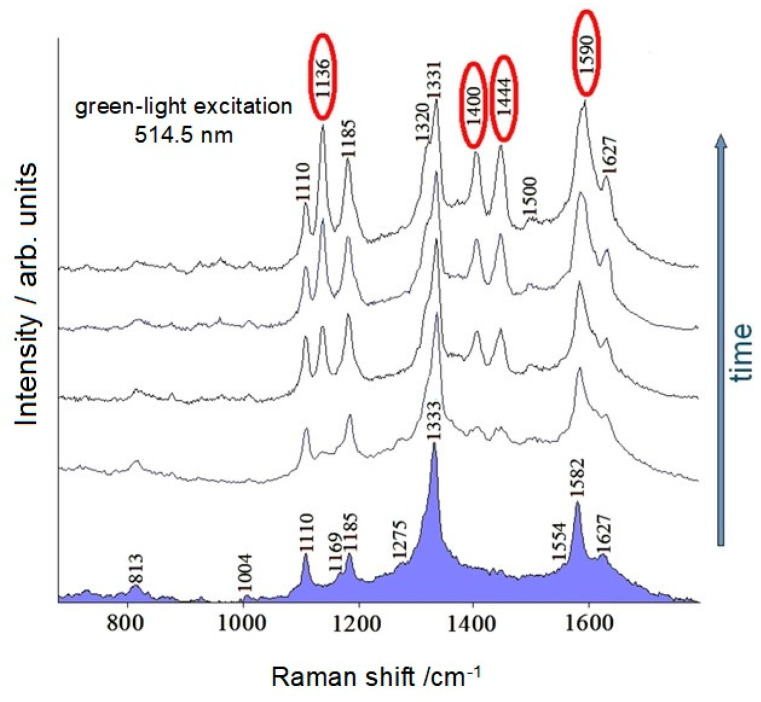
SERS spectra of DANS in Ag colloid, recorded every two minutes.

**Figure 13 molecules-28-00573-f013:**
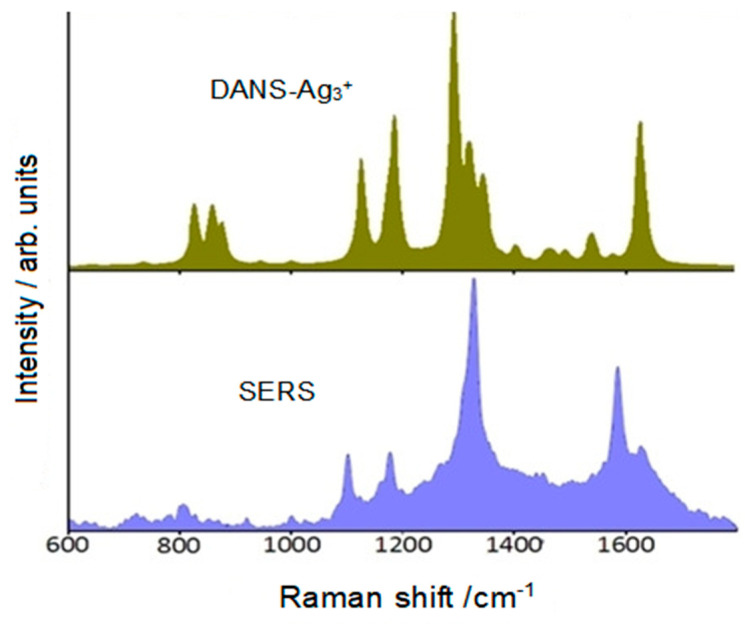
SERS spectrum of DANS and Raman spectra of DANS/Ag complex.

**Figure 14 molecules-28-00573-f014:**
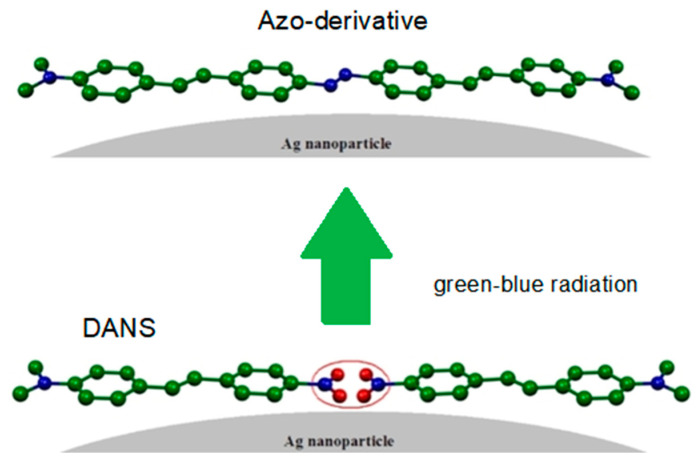
Adsorption and photoreaction of DANS on silver nanoparticles.

**Figure 15 molecules-28-00573-f015:**
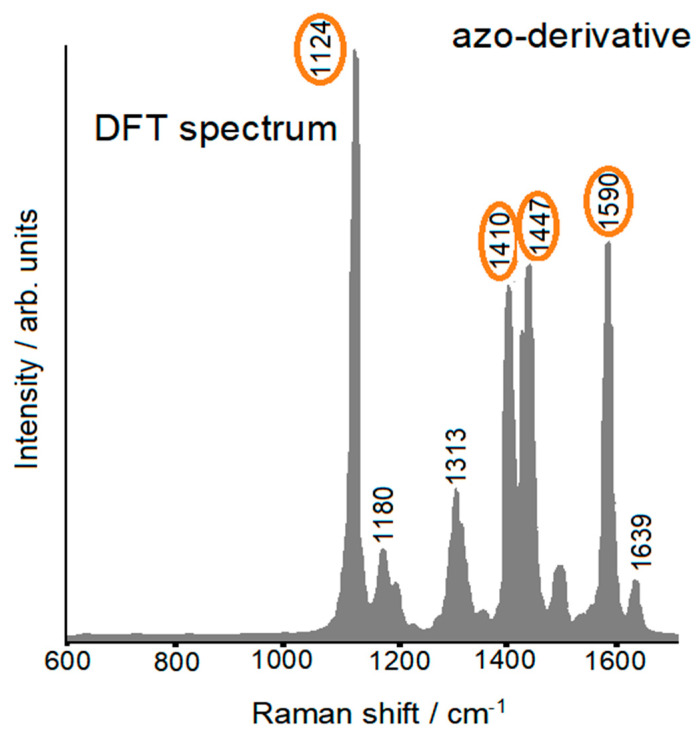
DFT-simulated Raman spectra of the proposed azo-derivative.

## Data Availability

Not applicable.

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
