# Peer review of "Can DFT Calculations Provide Useful Information for SERS Applications?"

_molecules, 2023, doi:10.3390/molecules28020573_

Round 1

Reviewer 1 Report

This work is of great significance, which can help researchers to further explore the mechanism of SERS and broaden the application of Raman spectroscopy.

1.     In recent decades, SERS spectroscopy has proved to be of fundamental importance in applications in a variety of research fields. In addition to the application fields in this article, there are more application fields that should be included and explained, such as food safety, medical diagnosis, environmental pollution, etc.

2.     Two N(9)H-B DFT results appeared in Fig. 2. Why?

3.     The SERS spectroscopy of 1,2,4-triazole on copper was showed in fig.9, and a DFT study was stated in this paper, but there is no DFT spectrum.

4.     DFT-simulated Raman spectra of the proposed azo-derivative was showed in fig.15, but there should be SERS spectrum of it so as to explain and verify the DFT results.

5.     The format specification should be carefully checked, eg, in 4.1 and 4.2 section.

Author Response

In the attached file I reply to the comments of Reviewer 1. Maurizio Muniz-Miranda

Reviewer 2 Report

The review presented by the authors seems interesting to me, as they give a contribution on the application in the SERS technique, which is normally essential when analyzing samples that could emit a lot of fluorescence.

I only have one observation in the "Conclusions" section, where the authors should more prominently display the contributions towards some priority health areas worldwide.

As for the format of the document, I leave these observations that could be addressed without much complexity:

1. Line 8, In the “Abstact” section, the word “Density” is in bold, it should be in normal format.

2. Line 261-270, in the “Sensors” section, the text has a different font than the rest of the text. Line 325-338, in the "Biomedicine" section.

3. Figs. 9, 11, 12, 13 and 15 should be with better resolution for the reader. In addition you must use the same size of dimensions for each image.

Author Response

In the attached file I reply to the comments of Reviewer 2. Maurizio Muniz-Miranda

Round 2

Reviewer 1 Report

The questions were fully answered.